# Antiviral Effect of Nonfunctionalized Gold Nanoparticles against Herpes Simplex Virus Type-1 (HSV-1) and Possible Contribution of Near-Field Interaction Mechanism

**DOI:** 10.3390/molecules26195960

**Published:** 2021-10-01

**Authors:** Edyta Paradowska, Mirosława Studzińska, Agnieszka Jabłońska, Valeri Lozovski, Natalia Rusinchuk, Iuliia Mukha, Nadiia Vitiuk, Zbigniew J. Leśnikowski

**Affiliations:** 1Institute of Medical Biology, Polish Academy of Sciences, 106 Lodowa St., 93-232 Łódź, Poland; eparadow@cbm.pan.pl (E.P.); mstudzinska@proteonpharma.com (M.S.); agnieszka.jablonska@poczta.onet.eu (A.J.); 2Institute of High Technologies, Taras Shevchenko National University of Kyiv, 64/13 Volodymyrska St., 01033 Kyiv, Ukraine; n.m.rusinchuk@knu.ua; 3Chuiko Institute of Surface Chemistry of National Academy of Sciences of Ukraine, 17 General Naumov St., 03164 Kyiv, Ukraine; iu.mukha@gmail.com (I.M.); n.vityuk@gmail.com (N.V.)

**Keywords:** antivirals, nanoparticles, HSV-1, near-field mechanism

## Abstract

The antiviral activity of nonfunctionalized gold nanoparticles (AuNPs) against herpes simplex virus type-1 (HSV-1) in vitro was revealed in this study. We found that AuNPs are capable of reducing the cytopathic effect (CPE) of HSV-1 in Vero cells in a dose- and time-dependent manner when used in pretreatment mode. The demonstrated antiviral activity was within the nontoxic concentration range of AuNPs. Interestingly, we noted that nanoparticles with smaller sizes reduced the CPE of HSV-1 more effectively than larger ones. The observed phenomenon can be tentatively explained by the near-field action of nanoparticles at the virus envelope. These results show that AuNPs can be considered as potential candidates for the treatment of HSV-1 infections.

## 1. Introduction

The strong ability of microorganisms and viruses to develop drug resistance is a continuous problem in chemotherapy. Viruses have numerous mechanisms of genetic variations for their survival [1,2,3,4]. The development of viral resistance necessitates new approaches to antiviral drug development. Therefore, improvements of new medicines and methods of antiviral therapy continue. Nanotechnologies provide numerous opportunities for new drug discovery. The fact that nonfunctionalized nanoparticles demonstrate antiviral activity [5,6,7,8,9] is of particular interest, especially because the observed activity probably results from a new mechanism of action. The known experimental facts about the antiviral action of nonfunctionalized nanoparticles [10] and new findings described in this communication allow for hypothesizing that a mechanism based on the near-field interaction between nanoparticles and the virus envelope and/or its components may take place.

HSV-1 and herpes simplex virus type-2 (HSV-2, genital herpes) are members of the human Herpesviridae family and the Alphaherpesvirinae subfamily. HSV-1 is one of the most common human infections worldwide, affecting 60–95% of the adult population worldwide [11]. Transmission of both HSV-1 and HSV-2 occurs during close personal contact. Most humans are infected with HSV-1 in childhood or early adolescence and remain latently infected throughout life. Sexual contact is the primary route of HSV-2 transmission. The use of antiviral agents after the establishment of latency will not result in the elimination of the virus although it helps to control the infection. All herpes viruses are enveloped virions with an icosahedral-shaped nucleocapsid containing the double-stranded linear DNA genome. Neurovirulence and latency of HSV have a direct impact on humans and the course of the disease. This, in turn, can result in profound disease and severe neurological sequelae, including HSV encephalitis. HSV evades the host’s immune system and establishes persistent infection. During latency, the HSV genome is maintained in the neuronal cells in a repressed state. The viral genome may subsequently become activated and transported via the neuron’s axon to the skin, resulting in viral replication and redevelopment of herpetic lesions. HSV incidence and severity have increased over the past decades owing to the increasing number of immunocompromised patients and sexual activities.

Several therapeutic options are available for HSV infections with the nucleoside analogs acyclovir (ACV), famciclovir (FCV), and valacyclovir (VCV), which target the viral DNA polymerase [12], in the first line. The guanosine analog ACV remains the gold standard in the treatment of herpes virus infections that exhibit both high selectivity and low toxicity [13]. Incorporation of the triphosphate form into the growing DNA chain by the viral polymerase in place of guanosine triphosphate leads to chain termination and inhibition of viral replication. ACV is used for the systemic treatment of HSV infections including genital and labial herpes. Besides ACV and its prodrug VCV, other nucleoside analogs, FCV and trifluridine, are also used for HSV treatment. However, incomplete suppressive treatment and resistance are serious disadvantages of these drugs. Currently, therapeutic vaccines for HSV-2 patients with genital herpes and the helicase-primase inhibitors are new specific anti-HSV drugs in development [14]. However, there is no drug available that can eliminate a latent infection, and the prolonged clinical use of antivirals in immunocompromised patients may lead to the incidence of treatment failure due to the development of antiviral-resistant virus strains [15].

Recent studies have shown that metal nanoparticles, particularly silver nanoparticles (AgNPs), can be effective against different types of viruses [16,17]. The possibility of applying metal-based nanoparticles, including silver, gold, tin, and zinc oxide, for the treatment of herpesvirus infections was also reported [18,19,20,21]. Among known methods for the treatment of HSV-caused infections, nanogold-based methods represent a promising direction. It was observed that AuNPs possess a high surface density of free electrons that results in inherent optical, electrical, and catalytic properties; as a result, these are widely researched as nanocarriers [22,23]. AuNPs are described as suitable for numerous biosensing functions and applications, including virus detection [24,25]. AuNPs functionalized with sulfonate ligand (MES) are nontoxic and effective against HSV-1 because of inhibition attachment of the virus to the cell surface [18]. The advantage of another, longer sulfonate linker (MOS) was also shown for AuNPs’ modification for their next multivalent binding with HSV-2 and further irreversible viral deactivation [26]. The virucidal mechanism for AuNPs modified with MOS compared to a virustatic found for the 2-(N-morpholino)-ethanesulfonic acid (MES) ligand was established. At the same time, no inhibitory activity of nonmodified citrate-coated nanoparticles was detected. Mechanistic studies using AuNPs capped with MES revealed that nanoparticles interfere with the attachment of HSV-1 to the host’s cell, viral entry, and cell-to-cell spread, thereby inhibiting viral infection [18]. In general, the main importance is attributed to spatially oriented functional groups anchored on nanoparticle surfaces for their next binding with the virus, and gold nanoparticles themselves serve as a carrier only. However, it has been shown that AuNPs, when unmodified with viral-specific molecules but stabilized with gallic acid, are also effective toward HSV-1 and HSV-2 in a dose-dependent manner [27]. The antiviral activity of nonmodified AuNPs against HSV-1 was demonstrated [28]. Hence, in the case of nonfunctionalized AuNPs, the physical and chemical properties of the AuNPs can be considered as the primary reason for their antiviral activity.

To explain nonfunctionalized AuNP antiviral activity based on the physical interaction of the virion with AuNPs, we postulate the contribution of a dispersion forces mechanism. The effect of electric fields induced by inhomogeneous distribution of electric charge around nanoparticles should be considered as the first possible cause and the driving force behind the AuNP antiviral activity. As such properties change with dimension, particle size should be the critical parameter for antiviral activity. Thus, the objective of the present study was to evaluate the effectiveness of unmodified citrate-capped gold nanoparticles against HSV-1 and to check the differences in the CPE of HSV-1 for nanoparticles of different sizes. In addition, we propose a physical mechanism of interaction between nanosized gold and the virus.

Most human viruses have a quasi-spherical shape with a linear dimension of about 100 nm. The characteristic dimensions of the nanoparticles are about few nanometers. Thus, we can use the model of a virus that is a spherically shelled solid nanoparticle whose core is characterized by a dielectric constant $\varepsilon_{vir}$ and whose shell is characterized by a dielectric constant $\varepsilon_{vir\;shell}$. The dielectric constant of the nanoparticle is $\varepsilon_{p}$. The domains of the high field and low field indicate the existence of gradients of the local field on the surface of the virus shell. The proteins and glycoproteins on the surface of the virus envelope contain the polar sites [29]. The dipole moments of the polar sites are under the action of the inhomogeneous field. The field may be caused, among others, by daylight, external light illumination, or it can be the field of vacuum fluctuations. It means that the forces acting on the viral surface proteins (ponderomotive forces) arise via $F=-P_{i}\cdot \partial E_{j}/\partial x_{i}$, where $P_{i}$ is the component of the dipolar moment, and $E_{j}$ is the local electric field at the virus envelope. This may block the interaction of viral proteins with cell receptors and penetration of the virus into the cell interior, resulting in an antiviral effect. Moreover, the long-term action of ponderomotive forces on the viral envelope can lead to the destruction of the envelope [6,9].

The effect of the local-field enhancement necessitates the existence of an external field. In normal conditions, the virus–nanoparticle system is situated under daylight action. When an additional field source acts on the system, one can expect some influence of the external light on the virus infection ability; such an additional lighting effect leading to an increase in the antiviral properties of nanoparticles has been observed [8].

Herein, we demonstrate that gold nanoparticles undergo a size-dependent interaction with the virus. Pretreatment of the virus with nanoparticles reduced the CPE of HSV-1 in a Vero cell culture. This phenomenon can be tentatively explained by the nanoparticles’ prevention of viral attachment, penetration into cells, and cell-to-cell spread.

## 2. Materials and Methods

### 2.1. Au Nanoparticles

#### 2.1.1. Gold Nanoparticles’ Preparation

Colloidal solutions of gold nanoparticles (AuNPs) were synthesized via chemical reduction of tetrachlorauric acid (HAuCl_4_, Merck, Darmstadt, Germany) with trisodium citrate (NaCit, Na_3_C_6_H_5_O_7_, Acros Organics, Geel, Belgium) in aqueous solutions according to the Turkevich method [30,31]. Thus, HAuCl_4_ was put into a boiling solution of NaCit used in double or quadruple molar excess, stirred while boiling for 5 min, then cooled at room temperature. A higher concentration of reductant leads to the formation of nanoparticles of a smaller size. The colloids were brought to their final concentrations: C_Au_ = 1.5 × 10^−4^ M and C_NaCitr_ = 9 × 10^−4^ M.

#### 2.1.2. The Particle Size Distribution

The AuNPs’ size distribution was studied by a laser correlation spectrometer Zeta Sizer Nano S (Malvern Panalytical Ltd., Malvern, UK) equipped with a correlator (Multi Computing Correlator Type 7032 CE) by dynamic light scattering (DLS). A helium–neon laser LGN-111(Polaron, Lviv, Ukraine) was used with an output power of 25 mW and wavelength of 633 nm to irradiate the suspension. The registration and statistical processing of the scattered laser light at angle 173° from the suspension were performed three times during 120 s at 25 °C. The resulting autocorrelation function was treated with standard computer programs PCS-Size mode v.1.61.

#### 2.1.3. The Absorption Spectroscopy

The absorption spectra of gold colloids were recorded in the UV-visible region by a spectrophotometer Lambda 35 (Perkin-Elmer, Norwalk, CT, USA) in 1 cm quartz cells.

### 2.2. Cells and Virus

Vero cells (ATCC CCL-81; American Type Culture Collection, Manassas, VA, USA) derived from the normal kidney epithelial cells of an African green monkey were cultured in Eagle’s minimum essential medium (EMEM; Sigma-Aldrich Co. Ltd., Ayrshire, UK). The growth medium was supplemented with 10% heat-inactivated fetal bovine serum (FBS, Sigma-Aldrich Co., St. Louis, MO, USA), 2 mM L-glutamine (Sigma-Aldrich Co., St. Louis, MO, USA), 100 units/mL penicillin G with 100 μg/mL streptomycin (Sigma-Aldrich Co., St. Louis, MO, USA), and amphotericin B 50 μg/mL (Sigma-Aldrich Co., St. Louis, MO, USA). The cells were cultured in a growth medium at 37 °C in a humidified 5% CO_2_ environment. The laboratory strain of HSV-1, strain MacIntyre (ATCC VR-539), was propagated and titrated in Vero cells in EMEM supplemented with 2% FBS. For propagation of the virus, confluent monolayers of Vero cells were infected with HSV at a multiplicity of infection (MOI) of 1 plaque-forming unit (PFU) per 100 cells (MOI = 0.01). After two to three days of infection, when the cytopathic effect was visible, the total virus was harvested and titrated by plaque assay.

### 2.3. Virus Titration

Titration of the viral loads in supernatants was performed by the method of Reed–Muench, and the titer was expressed per mL [32]. For the plaque reduction assay, Vero cells grown in 24-well plates (2 × 10^5^ cells/well) were inoculated with HSV-1 as described above. After 48 h of incubation at 37 °C (5% CO_2_), the cells were fixed with methanol for 15 min and stained with 0.05% methylene blue (Sigma-Aldrich) for 15 min. The HSV-1 plaques were counted under a microscope. Antiviral activity was expressed as the compound concentration required to reduce the number of viral plaques to 50% of the control (virus-infected but untreated).

For CPE, inhibitory assays were carried out in confluent Vero cell monolayers (2 × 10^4^ cells/well) growing in 96-well plates. The cell cultures were inoculated with HSV-1 as described above. After 48 h incubation at 37 °C in 5% CO_2_, the number of viable cells was determined by the MTT method. We used the Reed–Muench method to calculate TCID50. Antiviral activity was expressed as IC_50_ (50% inhibitory concentration), the concentration required to reduce virus-induced cytopathicity by 50% compared to the untreated control. Tenfold dilutions of freeze supernatants were utilized to inoculate Vero cell monolayers in 96-well plates, and infected cells were maintained in culture for 48 h at 37 °C in 5% CO_2_. Each sample was examined in triplicate.

### 2.4. Cytotoxicity Assay

To check that the AuNPs did not exert toxic effects on cells, the Vero cell monolayers were exposed to AuNP preparations. The number of viable cells was determined using the 3-(4,5-dimethylthiazol-2-yl)-2,5-diphenyltetrazolium bromide (MTT; Sigma-Aldrich Co., St. Louis, MO, USA) assay, which is based on the reduction of yellowish MTT to insoluble and dark blue formazan by viable cells. Vero cells were subcultured in 96-well plates at a seeding density of 2 × 10^4^ cells/well in EMEM supplemented with 10% FBS, L-glutamine, and antibiotics at 37 °C in a humidified 5% CO_2_ environment [33,34]. Confluent monolayers of cells were treated with preparations of AuNPs I and AuNPs II at concentrations of 0.295 and 5.9 μg/mL, respectively (six wells for each concentration). The stabilizer (sodium citrate, SC), AuNPs I, and AuNPs II were diluted 1:5 and 1:100 in EMEM supplemented with 2% FBS, 2 mM L-glutamine, and antibiotics (maintenance medium). The control consisted of Vero cells with no AuNPs and a stabilizer. After 48 h of incubation at 37 °C in 5% CO_2_, the number of viable cells was determined by adding MTT solution (5 mg/mL) to each well. The cells were incubated for a further 2 h at 37 °C in 5% CO_2_. Then, the formazan crystals were dissolved with dimethylsulfoxide (Sigma-Aldrich Co., St. Louis, MO, USA). The absorption values were measured at 550/670 nm using a microplate reader (Benchmark Plus, Bio-Rad Laboratories, Hercules, CA, USA). The reported data represent the percentage of cell viability compared with controls. Cytotoxicity of the compounds is expressed as the 50% cytotoxic concentration (CC_50_), which is the concentration required to reduce cell growth by 50% compared to untreated controls. Vero cell viability in each well is presented as a percentage of control cells. The CC_50_ was calculated by linear regression analysis of the dose-response curves obtained from the data.

### 2.5. Antiviral Assay

Different methods were used to treat the cell monolayers, to assess the effect of AuNPs on the inhibition of HSV-1 infectivity. The Vero cells were grown in 96-well (2 × 10^4^ cells/well) and 24-well plates (2 × 10^5^ cells/well) and exposed to a non-toxic concentration of AuNPs and HSV-1 infected. HSV-1 was titrated on a Vero cell line post-treatment with and without AuNPs. Antiviral activity was determined by the difference between the HSV-1 titers in untreated and treated cells.

#### 2.5.1. Virus Pretreatment Assay

To analyze AuNP influence on viral attachment to host cells, HSV-1 was pre-treated with AuNPs before Vero cell infection. First, the HSV-1 suspension was incubated in the presence of different concentrations of AuNPs for 0 min, 15 min, 1 h, and 4 h, and then added to confluent Vero cell monolayers at a MOI of 0.001 plaque-forming units (PFUs) per cell for 1 h at 37 °C. The treated cells were washed with phosphate-buffered saline (PBS), overlaid with fresh culture medium, and incubated for 48 h. The T0 time point means that AuNPs were added to the virus suspension and confluent Vero cell cultures immediately. Cells were incubated for 48 h at 37 °C in 5% CO_2_ and observed under an inverted microscope until typical CPE was visible. CPE was observed at 48 h post-infection (h p.i.). The supernatants from 24-well plates were collected, stored at −20 °C, or titrated as described before. Vero cells in 24-well plates were fixed with methanol, stained with 0.05% methylene blue, and washed with PBS, whereas 96-well plates were treated with MTT [33]. A negative control and virus control were included in each sample plate.

#### 2.5.2. Post-Treatment Assay

Vero cell monolayers were infected with HSV-1 at a concentration of MOI = 0.001 [26]. After a 1 h of adsorption at 37 °C, the virus inoculum was removed, and the cells were washed three times with PBS to remove the unattached virus. The AuNPs were then added to the inoculum in different dilutions of AuNPs at the following times: 0 h, 2, 4, and 24 h p.i. The T0 time point means that AuNPs and stabilizer were added to Vero cell cultures immediately after adsorption. The cell monolayers were incubated with the compounds for 48 h until the typical CPE was visible. As above, the supernatants after 48 h of incubation were collected, stored at −20 °C, and Vero cells in 24-well plates were fixed with methanol and stained with methylene blue, whereas 96-well plates were treated with MTT.

### 2.6. Cryogenic Transmission Electron Microscopy

Cryogenic Transmission Electron Microscopy images were obtained using a Tecnai F20 TWIN microscope (FEI Company, Hillsboro, OR, USA) equipped with a field emission gun, operating at an acceleration voltage of 200 kV. Images were recorded on an Eagle 4k HS camera (FEI Company, Hillsboro, OR, USA) and processed with TIA software (FEI Company, Hillsboro, OR, USA). Specimen preparation was done by vitrification of the aqueous solutions on grids with holey carbon film (Quantifoil R 2/2; Quantifoil Micro Tools GmbH, Jena, Germany). Before use, the grids were activated for 15 s in oxygen plasma using a Femto plasma cleaner (Diener Electronic, Ebhausen, Germany). Cryo samples were prepared by applying a droplet (3 µL) of the solution to the grid, blotting with filter paper, and rapid freezing in liquid ethane using a fully automated blotting device Vitrobot Mark IV (FEI Company, Hillsboro, OR, USA). After preparation, the vitrified specimens were kept under liquid nitrogen until they were inserted into a cryo-TEM holder Gatan 626 (Gatan Inc., Pleasanton, CA USA) and analyzed in the TEM at −178 °C. The Cryo-TEM measurements were performed under a contractual service agreement with CMPW PAN in Zabrze, Poland.

### 2.7. Statistical Analysis

Statistics including the mean and standard deviation (SD) were analyzed with the GraphPad Prism software using a non-parametric unpaired *t*-test. A *p*-value of ≤0.05 was considered significant. The data were obtained from two or three independent experiments.

## 3. Results and Discussion

### 3.1. Characterization of AuNPs

Following calculations discussed in Section 3.3. below, which showed that the highest energy of interaction between the nanoparticle and the virus surface could be expected for the particle size range 5–15 nm, we chose particles of 10 nm (AuNPs I) and 16 nm (AuNPs II) in our study.

We prepared samples of AuNP colloids that differed in size according to DLS measurements, namely, on a “number” basis: AuNPs I had an average size of 10 nm, and AuNPs II were 16 nm (Figure 1). The AuNPs had a typical band of localized surface plasmon resonance in UV-Vis spectra with a maximum at 520 nm and carried a negative charge with zeta potential values of −29 mV (AuNPs I) and −42 mV (AuNPs II).

For all the experiments below, AuNPs were dissolved in a maintenance medium and used at concentrations of 0.295 and 5.9 μg/mL. We decided to test AuNP antiviral activity at a concentration not higher than 5.9 μg/mL to avoid potential cytotoxicity problems and not lower than 0.295 μg/mL because of the observed moderate antiviral activity of the AuNPs under the conditions studied.

### 3.2. Nanoparticle Cytotoxicity

To rule out the possibility that the reduction of infectivity was caused by the cellular toxicity of AuNPs, monolayers of Vero cells were incubated with different concentrations (0.295 and 5.9 μg/mL) of each type of AuNP for different time points. The MTT results revealed that AuNP II and the stabilizer did not induce cell death after 48 h incubation at concentrations of up to 5.9 and 62 μg/mL, respectively. Data are representative of three independent experiments, and values are expressed in mean ± SD.

The smaller-sized AuNPs I (10 nm) exhibited stronger toxic effects than AuNPs II (16 nm) in the Vero cell cultures. In addition, the toxic effect depended on the concentrations of gold nanoparticles (Figure 2). The percentages of viable cells relative to the control cultures were ca. 71% (AuNP I) and 99% (AuNP II) at 0.295 μg/mL, and 58 and 93%, respectively, at 5.9 μg/mL. The results revealed that cell viability was maintained close to 100% for sodium citrate as a stabilizer.

### 3.3. Nanoparticle Adsorption on the Virus

The first stage of interaction between the virus and the nanoparticles is their physical interaction via dispersion forces. This means that owing to fluctuation, the inhomogeneous distribution of electric charge (nanoparticle polarization) arises. This immediately induces the electric field at the opposite object (the other nanoparticles in the case of Van der Waals interaction for the virus in the studied case), which is the reason for the inhomogeneous distribution of electric charge in the other particle (the virus). The interaction between the dipole moments leads to the Van der Waals forces. The reasons for the polarization fluctuation can be thermal fluctuations, the electric field of vacuum fluctuations, external electric field, and other factors [35].

Let us consider the spherical Au nanoparticle located near the HSV-I virion. The virion has glycoprotein spikes on its surface [36], and so its uneven surface should be considered. The virion has an icosahedral shape and a much bigger size compared to the nanoparticle. The abovementioned factors allow for simulating the studied system as the spherical homogeneous nanoparticle located close to the nanostructured surface (Figure 3). Considering the nanoparticle adsorption on the virus surface, we considered different locations of the nanoparticle (see Figure 3, cases 1–3) and calculated the adsorption potential between the nanoparticle and the surface.

The interaction potential in the system is presented in the following form, as in [35]:(1)$$U(z)=F\left(P\left(d\right)\right)-F\left(P\left(d=\infty \right)\right)$$
where d is the distance between the nanoparticle center and the top of the virus spike (see Figure 3c) and $F\left(P\right)$ is the free energy of the system, which is described as in [37]:(2)$$F\left(P_{i}\right)=U_{0}+\frac{1}{2\alpha }P_{i}^{2}+\frac{1}{4}\beta P_{i}^{4}-E_{i}\left(P_{i}\right)\cdot P_{i}.$$

We need to find the ground state of the system, which means the state with the energy minimum. To find it, we used the Green’s function method, which allowed us to present the local field $E_{i}\left(P_{i}\right)$ via the polarization of the nanoparticle $P_{i}$ and the Green’s function of the medium in which the nanoparticle is located:(3)$$E_{i}\left(P_{j}\right)=E_{i}^{\left(0\right)}-k_{0}^{2}\cdot G_{ij}\left(R,R’,\omega \right)\cdot P_{j}.$$

Consequently, Equation (2) can be differentiated by the nanoparticle polarization, which in turn, leads to the definition of the polarization corresponding to the ground state of the system, such as in [38]. Thus, for the calculations, we need to find the Green’s function of the system in which the nanoparticle is located. In the studied case, the system comprises two half-spaces with an uneven interface. The Green’s function of such a system can be found using the pseudo-vacuum Green’s function method [39] as in [40]. It should be noted that the interface has many spikes and considering the effect of all these spikes makes the calculations too complicated. It was shown that the effect of the spikes located far from the nanoparticle is much smaller compared to the effect of the closer ones, so it was omitted [40,41].

Hence, the modeling should be constructed in the following stages:

1. Construction of a model of the virus surface with spikes based on experimental studies of the virus structure.

2. Estimation of the “critical” distance between two spikes based on the effective susceptibility concept and pseudo-vacuum Green’s function, as was conducted in [40,41]. In this case, we considered the curved surface with R >> r (R is the virus radius, r is the radius of the spike base) with cylinder spikes on it. These cylinders are the model of the virus spikes. Here, the “critical” distance means that the term in the expression for the effective susceptibility of spike 0 caused by spike 1 located at this “critical” distance is more than 100 times less than other terms.

3. Simplification of the model by elimination of all the spikes located farther than the “critical” distance for simplification of the calculations. Calculation of the adsorption potential of the nanoparticle in different locations is made using this model, as in [40].

For each of the three described cases, we studied the influence of spikes. It was concluded that similar to results in [40], for cases 1 and 3 only, three spikes should be considered: the central one and one from each side. For case 2, four spikes should be considered: two spikes from each side of the nanoparticle in one dimension.

For the gold nanoparticles the model of the core–shell spherical nanoparticle was used. The core is gold, the material of nanoparticles. The shell is the stabilizer, which may be described as a thin shell around the gold core. In the calculations we assumed that the shell was homogeneous. The dielectric constant of gold in the calculations was equal to −10.5 + 1.4 *i* [42]. It should be noted that all interactions were considered in the presence of visible light, so the values were used for the visible light range. The nanoparticle shell was variable as it was formed by the stabilizer molecules; its thickness depends much on the nanoparticle size, stabilizer concentration, and material. For the trisodium citrate the shell thickness was around 0.4–0.7 nm [43], and for the calculations we used the shell thickness value of 0.5 nm and the shell dielectric constant value of 1.3 as in [44].

The virus was described as a spherical structure with a nonhomogeneous shell; the surface had some cylindric spikes (mainly glycoproteins). The radius of the inner part of the virion in the calculations was equal to 60 nm, and the whole shell thickness was taken as 20 nm: 10 nm of the homogeneous layer and 10 nm the height of the cylinders [45]. As the dielectric properties were not known exactly for the viruses, we chose the ones for DNA [46], which was the inner part, and viral proteins and glycoproteins [47], which corresponded to the shells.

The results of calculations showed the energy of the interaction between the nanoparticle and the virus surface depending on the distance between them. As shown in Figure 3c the distance “d” changed, but the relative position of the nanoparticle center and the edge of the spike were stable. The minimum of the potential indicated the physical adsorption of the nanoparticle on the virus surface, whereas the depth of the minimum indicated the energy of adsorption. The case with the deepest energy minimum was the most energetically favorable state. Consequently, comparing the potentials for the different relative locations of the virus center and the spike edge, it was seen that the deepest energy minimum and the closest position of the equilibrium system state were for case 1. Similar results were observed for all the nanoparticle sizes considered. However, for the 20 nm nanoparticle these changes were not so obvious. Comparing the potentials for different nanoparticle sizes it could be seen that the deepest minimum was for the smallest nanoparticles. Hence, it may be supposed that the antiviral effect was higher for the smaller nanoparticles, which was indeed observed in our work.

From the results of calculations (Figure 4) of the adsorption potential of the systems with different nanoparticle locations, it can be also stated that the nanoparticle adsorption to the virus spike was the most energy-efficient state. This means that the nanoparticle did not penetrate between the spikes and did not get closer to the virus envelope. Hence, based on the calculation results, it can be hypothesized that the nanoparticles were adsorbed on the virus surface uniformly according to their spikes. This may disturb the viral attachment to the cellular receptors and prevent entry to the cell or fusion with its membrane. The process of the nanoparticle adsorption to the HSV-I virion was studied by Cryo-TEM (Figure 5). It was seen that nanoparticles likely adsorbed to the virus spikes, as was described previously, which indicated that this process was possibly caused by a dispersion interaction between the nanoparticle and the virus. However, an experimental confirmation on the molecular level that nanoparticle adsorption to the virus spike is the most energy-efficient state was a challenge that was not addressed in this preliminary communication.

### 3.4. Antiviral Effect of Nanoparticles

Different methods were used to treat the cell monolayers to assess the effect of AuNPs on the inhibition of HSV-1infection: (1) pretreatment assay in which the different AuNPs and HSV-1 were added to the confluent monolayer of Vero cells during or after viral adsorption, and (2) post-treatment assay in which Vero cell monolayers were first infected with HSV-1, and AuNPs were then added to the inoculum at different times. For all treatments, cells infected with HSV-1 were then incubated for 48 h at 37 °C.

To test whether AuNPs affected HSV-1 infectivity in vitro, we used a pretreatment assay. HSV-1 suspension was incubated with AuNPs for 0 min, 15 min, 1 h, and 4 h, and then added to the Vero cell monolayers.

The dose-dependent efficiency of nanoparticles on viral titers released into cell-free culture supernatants was observed. As shown in Table 1, incubation of virions with AuNPs I reduced the virus replication in a dose-dependent manner. A higher level of inhibition was observed with nanoparticles of 5.9 µg/mL incubated with HSV-1 at 1 or 4 h compared to nanoparticles at a concentration of 0.295 µg/mL. Four-hour pretreatment of the smaller-sized nanoparticles with the virus achieved up to a 100-fold decrease of the HSV-1 load, while bigger-size nanoparticles reduced the viral load twofold, at best, compared to infection control.

The viral loads in supernatants collected after 48 h p.i. were determined in confluent Vero cultures and calculated with the Reed–Muench method. AuNPs I pretreated with HSV-1 for 1 or 4 h showed a major antiviral effect at a concentration of 5.9 μg/mL (Table 1). AuNPs I caused up to a 100-fold inhibition of exogenous virus loads.

Because inhibition of viral infectivity could be a consequence of the action of AuNPs inside the cell at a post-entry event, we performed a post-treatment test by adding the nanoparticles at two concentrations 0–24 h after viral infection. Post-treatment did not have any measurable effect at the concentrations used, so the nanoparticles were not able to reduce the HSV-1 loads (data not shown).

Depending on the cell type there are two main pathways—non-endocytic and endocytic—of HSV entry into host cells and viruses use a similar set of viral surface glycoproteins to enter host cells. HSV attaches through the envelope glycoproteins to receptors on the surface of the host cell. This interaction allows for tight anchoring of the virion particle to the plasma membrane of the host cell and eventually leads to membrane fusion and virus penetration into the host cell. Gold nanoparticles probably block the interaction of the virus with the cell, which might be dependent on the nanoparticle size. The size of the AuNPs may determine the host–pathogen interaction, and smaller nanoparticles may cause higher binding efficiency. It was found that smaller-sized silver nanoparticles (AgNPs) attach to HSV-1, inhibiting the virus from attaching to host cells and ultimately resulting in attenuation of viral replication [48,49]. Halder et al. [27] synthesized gallic acid-stabilized mono-dispersed gold nanoparticles and found that they strongly inhibited the proliferation of HSV-1 infection in Vero cells. It is also possible that gold nanoparticles undergo a size-dependent interaction with HSV-1. These results revealed that AuNPs were capable of controlling viral infectivity, most likely by blocking the interaction of the virus with the cell, which might depend upon the size of the AgNP. Our results revealed that smaller gold nanoparticles with a size of 10 nm had better antiviral activity, although they showed increased toxicity. NP toxicity strongly depends on their physical and chemical properties, including size and shape. It was observed that different human cell types were more sensitive to small gold particles (1.4 nm) than gold particles 15 nm in size [50]. It is possible that smaller-sized AuNPs attach more easily to the viral envelope, resulting in the reduction of HSV-1 replication.

Electric field interaction with unmodified gold nanoparticles, or “physical” interaction, could be the first step before the tight “chemical” binding of AuNPs with the viral envelope through spikes built from glycoproteins, namely, through donor–acceptor or covalent bond formation between Au and peptide moieties.

## 4. Conclusions

1. No cytotoxicity of AuNPs II (16 nm) was observed in the Vero cell line up to a gold concentration of 5.9 μg/mL, while AuNPs I (10 nm) presented a greater cytotoxic effect.

2. Pretreatment, as well as post-treatment, of HSV-1 with AuNPs did not show a significant effect on the Vero cell viability. The reduction in CPE of HSV-1 after four-hour pretreatment of the virus with AuNPs II reached a maximum of 10%.

3. Smaller-sized nanoparticles were able to inhibit the HSV-1 replication in a pretreatment assay. A virus pretreatment assay showed that AuNPs reduced HSV-1 replication in a dose- and time-dependent manner.

4. It may be hypothesized that AuNP adsorption on the virion may disturb the virus attachment to the cellular receptors and prevent entry to the cell or fusion with its membrane.

5. AuNP adsorption to the virion spikes can be explained by their Van der Waals interaction.

## 5. Outlook

Gold nanoparticles with an average size of 10 nm demonstrated inhibitory activities against HSV-1 in a dose- and time-dependent manner at non-cytotoxic concentrations.

## Figures and Tables

**Figure 1 molecules-26-05960-f001:**
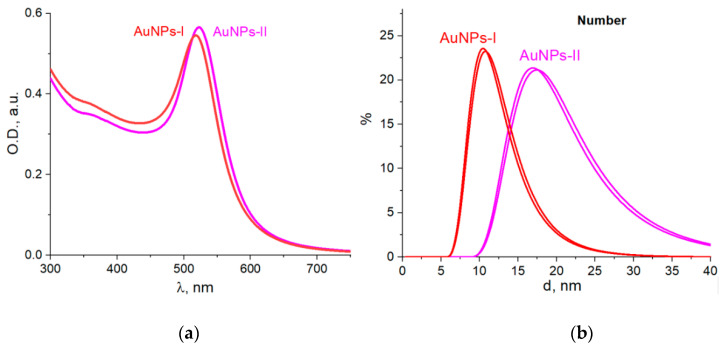
Absorption spectra (**a**) and DLS size distribution (**b**) of gold nanoparticles in obtained colloids.

**Figure 2 molecules-26-05960-f002:**
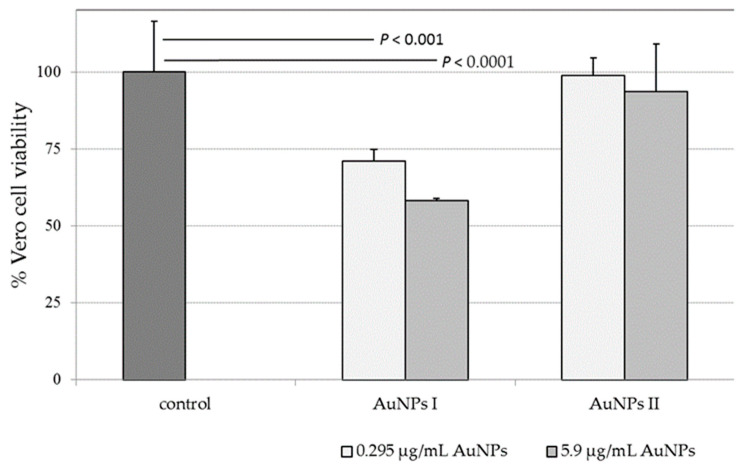
Cytotoxicity of AuNPs in Vero cells. Number of viable cells to the untreated cell control (%) after incubation for 48 h. AuNPs of approximate diameters of 10 nm (AuNPs I) and 16 nm (AuNPs II) were used at concentrations of 0.295 and 5.9 μg/mL, respectively. The cell viability was measured using an MTT assay. The data are shown as means ± SD of three experiments performed six times. *P,* Student’s *t*-test.

**Figure 3 molecules-26-05960-f003:**
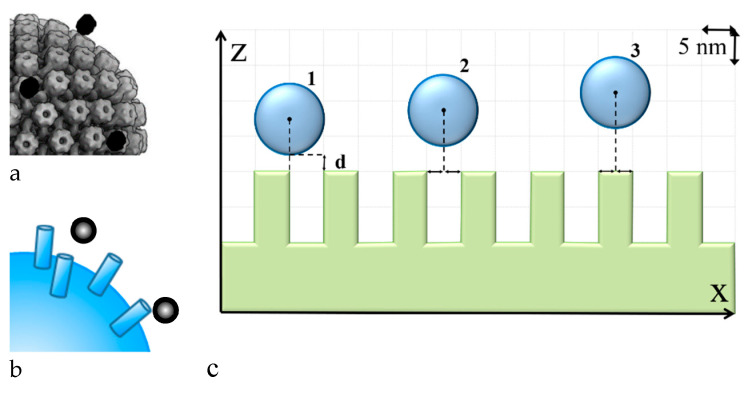
Sketch of the studied system. (**a**) is an idealized view of the virus surface with adsorbed nanoparticles; (**b**) is a model for calculations of adsorption potential for different sites of nanoparticles relative to virus spikes; (**c**) is the cross-section of ‘b’.

**Figure 4 molecules-26-05960-f004:**
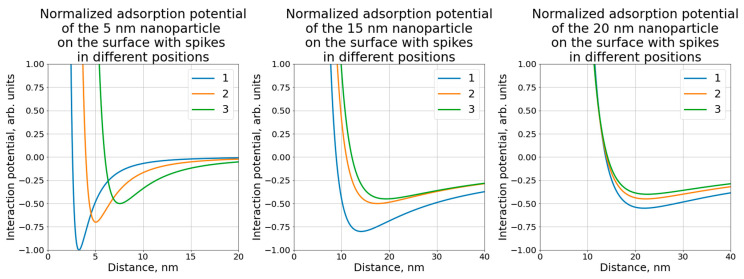
Interaction potential between the nanoparticle and the virus depending on the nanoparticle location near the virus spike (curve 1 corresponds to site 1 in Figure 3; curve 2 corresponds to site 2 in Figure 3; curve 3 corresponds to site 3 in Figure 3). All the results were normalized to the deepest energy minimum corresponding to case ‘1′ of 5 nm nanoparticles.

**Figure 5 molecules-26-05960-f005:**
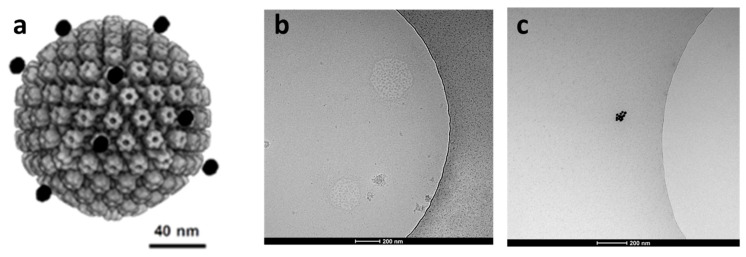
(**a**): Hypothetical interaction of Au nanoparticles ● and HSV-1 virion (HSV-1 model adapted from [36]). (**b**): Cryo-TEM of HSV-1 treated with AuNP 10 nm particles (1 h incubation time), (**c**): AuNP 10 nm particles.

**Table 1 molecules-26-05960-t001:** Pretreatment assay. Vero monolayers were treated AuNPs infected with HSV-1 at different times, and supernatants were titrated on virus replication.

AuNPs (Diameter)	Concentration (μg/mL)	HSV-1 Load in Vero Cell Supernatants (TCID_50_/mL)			
		After Pretreatment			Non-Treated with AuNPs
		T 0.25 h	T 1 h	T 4 h	
AuNPs I (10 nm)	0.295	4.30 × 10^5^ ±1.70 × 10^5^	1.02 × 10^6^ ±1.02 × 10^6^	4.30 × 10^5^ ±1.70 × 10^5^	5.50 × 10^6^
	5.9	4.26 × 10^5^ ±1.75 × 10^5^	5.50 × 10^4^	4.30 × 10^4^ ±1.70 × 10^4^	5.50 × 10^6^
AuNPs II (16 nm)	0.295	3.09 × 10^6^	5.50 × 10^6^	3.09 × 10^6^	5.50 × 10^6^
	5.9	2.42 × 10^6^ ±9.55 × 10^5^	3.09 × 10^6^	2.38 × 10^6^ ±9.05 × 10^5^	5.50 × 10^6^

Data are represented as mean ± SD of two experiments performed in triplicate. Abbreviations: T, time of pretreatment (h, hours); AuNPs I, size 10 nm; AuNPs II, size 16 nm. Samples from the supernatant medium were collected at 24 h p.i.

## Data Availability

Not applicable.

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
