# Peer review of "Antiviral Effect of Nonfunctionalized Gold Nanoparticles against Herpes Simplex Virus Type-1 (HSV-1) and Possible Contribution of Near-Field Interaction Mechanism"

_molecules, 2021, doi:10.3390/molecules26195960_

Round 1

Reviewer 1 Report

Comments on the manuscript ID1376076 by Paradowska et al. entitled “Antiviral effect of non-functionalized gold nanoparticles against herpes simplex virus type-1 (HSV-1) and possible contribution of near-field interaction mechanism”:

The authors report an experimental (in vitro) study of the antiviral activity of non-functionalized gold nanoparticles (AuNPs) against herpes simplex virus type-1 (HSV-1). In general, the article is well written, full of content material and beneficial information, but it requires improvement in the presentation/discussion of the results. Some shortcomings are listed below:

  • Title: to be shortened, the interaction mechanisms are not fully elucidated;
  • On page 3 the authors say that “This means that additional illumination by the monochromatic light with a resonant wave length may intensify the antiviral action.” The effect of the light on the HSV-1 antiviral activity of AuNPs is not followed in this study;
  • The authors conclude that the AuNPs undergo a size-dependent interaction with the virus, but only 2 samples of AuNPs’ colloids are prepared: AuNPs I, average size of 10 nm, and AuNPs II, average size of 16 nm;
  • There is no explanation why AuNPs are used only at concentrations 295 μg/mL and 5.9 μg/mL (twenty-times higher).
  • Section 3.3: the authors explain how the adsorption potential between the nanoparticle and the surface is calculated/modelled, but there is no data presented for the considered models of the systems with different nanoparticle locations. “From the results of calculations of the adsorption potential of the systems with different nanoparticle locations (0), it can be stated that the nanoparticle adsorption to the virus spike is the most energy-efficient state.” – in fact, no such results are listed;
  • Figure 4 is not of publication quality.

Author Response

Reviewer 1

  • Title: to be shortened, the interaction mechanisms are not fully elucidated;

We agree with the referee’s observation that the AuNPs/HSV-1 interaction mechanism was not fully elucidated in our case. For this reason, we use the cautious term "possible" in the title of our work. Nevertheless, due to the reasonable probability of the participation of near field  mechanism in the observed antiviral activity of gold nanoparticles (suggested also by others e.g. Mater. Adv., 2021, 2, 2188 in the case of SARS-CoV-2), we would like to keep this title. The more so that proposing this mechanism is one of the novelties of our work.

  • On page 3 the authors say that “This means that additional illumination by the monochromatic light with a resonant wave length may intensify the antiviral action.” The effect of the light on the HSV-1 antiviral activity of AuNPs is not followed in this study;

We appreciate the referee’s comment on this issue. Indeed, the effect of light on AuNPs antiviral activity has not been the subject of our research. The questioned sentence has been removed.

  • The authors conclude that the AuNPs undergo a size-dependent interaction with the virus, but only 2 samples of AuNPs’ colloids are prepared: AuNPs I, average size of 10 nm, and AuNPs II, average size of 16 nm;

Following calculations discussed in the manuscript and in answer to the reviewer 1 penultimate question which shows that the highest energy of interaction between the nanoparticle and the virus surface can be expected for the particle size range 5-15 nm we chosen particles of the size 10 and 16 nm in our study.

  • There is no explanation why AuNPs’ are used only at concentrations 295 μg/mL and 5.9 μg/mL (twenty-times higher).

We decided to test AuNPs antiviral activity at a concentration not higher  than 5.9 μg/mL to avoid potential cytotoxicity problems and not lower than 0.295 mg/mL due to observed moderate antiviral activity of the AuNPs under conditions studied.

  • Section 3.3: the authors explain how the adsorption potential between the nanoparticle and the surface is calculated/modelled, but there is no data presented for the considered models of the systems with different nanoparticle locations. “From the results of calculations of the adsorption potential of the systems with different nanoparticle locations (0), it can be stated that the nanoparticle adsorption to the virus spike is the most energy-efficient state.” – in fact, no such results are listed;

For the gold nanoparticles we use the model of the core-shell spherical nanoparticle. The core is gold, the material of nanoparticles. The shell is the stabilizer which may be described as a thin shell around the gold core. In the calculations we assumed that the shell is a homogeneous one. The dielectric constant of gold in the calculations was equal to -10.5 + 1.4 i [Refractive index database https//refractiveindexinfo Accessed November 20]. It should be noted that all interactions are considered in the presence of visible light, so the values were used for the visible range. The nanoparticles shell is variable as it is formed by the stabilizer molecules, its thickness depends much on the nanoparticles size, stabilizer concentration and material. For the trisodium citrate the shell thickness is around 0.4-0.7 nm [https://doi.org/10.1039/C7RA10759E], for the calculations we used the shell thickness value of 0.5 nm and the shell dielectric constant value of 1.3 as in [https://doi.org/10.1016/j.colsurfa.2010.08.035].

The virus was described as a spherical structure with a nonhomogeneous shell, the surface has some cylindric spikes (mainly glycoproteins). The radius of the inner part of the virion in the calculations is equal to 60 nm, and the whole shell thickness was taken 20 nm: 10 nm of the homogeneous layer and 10 nm is the height of the cylinders [https://doi.org/10.1111/j.1699-0463.1994.tb04882.x]. As the dielectric properties are not known exactly for the viruses, we chose the ones for DNA [https://doi.org/10.1049/iet-nbt.2008.0014], which is the inner part, and viral proteins and glycoproteins [http://dx.doi.org/10.1016/S0304-4165(03)00118-1], which corresponds to the shells.

The results of calculations show the energy of interaction between the nanoparticle and the virus surface depending on the distance between them. As it was shown in Fig. 3c the distance “d” changes but the relative position of the nanoparticle center and the edge of the spike are stable. The minimum of the potential indicates the physical adsorption of the nanoparticle on the surface, whereas the depth of the minimum indicates the energy of adsorption. The case with the deepest energy minimum is the most energetically favorable state. Consequently, comparing the potentials for different relative locations of the virus center and the spike edge, it can be seen that the deepest energy minimum and the closest position of the equilibrium system state is for the case ‘1’. Similar results may be observed for all the nanoparticles sizes considered. However, for the 20 nm nanoparticle these changes are not so obvious. Comparing potential for different sizes of the nanoparticles it can be seen that the deepest minimum is for the smallest nanoparticles. Hence, it may be supposed that the antiviral effect may be higher for the smaller nanoparticles, which was indeed observed in our work. However, an experimental confirmation on the molecular level that “nanoparticle adsorption to the virus spike is the most energy-efficient state” is a challenge that was not addressed in this preliminary communication.

Graphics for Fig. 4 available in the main body of the paper. 

Figure 4. Interaction potential between the nanoparticle and the virus depending on the nanoparticle location near the virus spike (curve 1 corresponds to site ´1´ in Fig. 3; curve 2 corresponds to site 2 in Fig. 3; curve 3 corresponds to site 3 in Fig. 3 ). All the results were normalized to the deepest energy minimum corresponding to case ‘1’ of 5 nm nanoparticles.

  • Figure 4 is not of publication quality.

The quality of Fig. 4 was improved.

Reviewer 2 Report

In this study, non-functionalized gold nanoparticles is used as materials to study its antiviral activity against herpes simplex virus type-1 (HSV-1) in vitro and explain preliminary action mechanism from physical point of view with novelty and innovation. At the same time, the steps of experiment have legible logic and rigorous argument. In a word, the research is highly original and recommended to receive after modification as follows.

  1. Characterization of AuNPs is so simple. Please add zeta potential and transmission electron microscope (TEM) images of AuNPs in Figure 1.
  2. In Figure, 0, 295 and 5,9 should be changed into “ 295”and “5.9”.
  3. Table 4 should be changed into “Table 1”. In addition, Please modify it in three-line table format.
  4. Conclusion 1 is incorrect. According to Figure 2, significant cytotoxicity of AuNPs (P<0.001) could be observed in the Vero cell line at the gold concentration of 295 and 5.9 μg/mLfor 10 nm. In addition, please add statistical analysis in the caption Figure 2.

Author Response

  1. Characterization of AuNPs is so simple. Please add zeta potential and transmission electron microscope (TEM) images of AuNPs in Figure 1.

The zeta potential values of -29 mV (AuNPs I) and -42 mV (AuNPs II) are provided in the text, just above Fig. 1 (Line 269).

Cryogenic transmission electron microscopy (TEM) experiments were performed only with AuNPs I, 10 nm particles that showed higher antiviral activity. The cryo-TEM image of these nanoparticles was included in Figure 5. The figure’s caption was changed accordingly “5. Left: Hypothetical interaction of Au nanoparticles ● and HSV-1 virion (HSV-1 model adapted from [32]). In the middle: Cryo-TEM of HSV-1 treated with AuNPs 10 nm particles (1 h incubation time),  right: AuNPs 10 nm particles.

  1. In Figure, 0, 295 and 5,9 should be changed into “ 295”and “5.9”.

We appreciate the reviewer’s important comment. We agree with the reviewer’s suggestion that in Figure 2 the decimal comma should be replaced by decimal point in the used AuNPs concentrations. We would like to note that the concentration was 0.295, not 295 μg/mL. According to the reviewer’s suggestion,  the concentration of the 5,9 μg/mL has been replaced with the 5.9 μg/mL in Figure 2. In addition, the concentration of the 0,295 μg/mL has been replaced with the 0.295 μg/mL. Please see page 7 in the Results section.

  1. Table 4 should be changed into “Table 1”. In addition, Please modify it in three-line table format.

The table was changed according to the referee’s request.

  1. Conclusion 1 is incorrect. According to Figure 2, significant cytotoxicity of AuNPs (P<0.001) could be observed in the Vero cell line at the gold concentration of 295 and 5.9 μg/mLfor 10 nm. In addition, please add statistical analysis in the caption Figure 2.

We appreciate the reviewer’s important and valuable comment. We agree with the opinion that the significant cytotoxicity of AuNPs  I (10 nm ) has been observed in the Vero cell line at the AuNPs concentration of 0.295 and 5.9 μg/mL. This information has been unintentionally omitted in the conclusions. In addition, we agree that the statistical analysis/test should be included in the caption of Figure 2.

According to the Reviewer’s suggestion,  the sentence “No cytotoxicity of AuNPs was observed in the Vero cell line up to a gold concentration of 5.9 μg/mL” has been changed to “No cytotoxicity of AuNPs II (16 nm) was observed in the Vero cell line up to a gold concentration of 5.9 μg/mL, while AuNPs I (10 nm) presented a greater cytotoxic effect”. Please see point 1 in the Conclusions. In addition, the phrase “P, Student’s t-test” has been included into a caption under Figure 2. Please see page 7 in the Manuscript.

In addition

The part “c” of Fig. 3 was replaced with a picture of higher quality.

Printing errors in the list of references have been corrected.

Round 2

Reviewer 1 Report

The detailed explanations given by the authors (Author's Notes 3-5) should be included in the text of the article. Figure 4 still needs improvement (text not readable).

Author Response

Reviewer 1 (Round 2)

The detailed explanations given by the authors (Author's Notes 3-5) should be included in the text of the article.

Note 3 (with necessary adjustements) has been included in the text in the front of the section 3.1 (p. 6, lines 262-265, line numbering as in revised version): “Following calculations discussed in section 3.3 below which show that the highest energy of interaction between the nanoparticle and the virus surface can be expected for the particle size range 5-15 nm (Fig. 4) we chosen particles of the size 10 nm (AuNPs I) and 16 nm (AuNPs II) in our study”

The sentence “The AuNPs’ preparations with different sizes were used for further tests (average size 10 nm AuNPs I and 16 nm AuNPs II)” located originally just below Figure 1 was deleted.

Note 4 has been included in the text on page 6 (lines 276-279), at the bottom of section 3.1: “We decided to test AuNPs antiviral activity at a concentration not higher  than 5.9 μg/mL to avoid potential cytotoxicity problems and not lower than 0.295 mg/mL due to observed moderate antiviral activity of the AuNPs under conditions studied”.

Note 5 has been included in the text on page 9 (lines 359-392) together with additional references 38-43, the following references have been renumbered accordingly: “For the gold nanoparticles the model of the core-shell spherical nanoparticle was used. The core is gold, the material of nanoparticles. The shell is the stabilizer which may be described as a thin shell around the gold core. In the calculations we assumed that the shell is homogeneous. The dielectric constant of gold in the calculations was equal to -10.5 + 1.4 i [38]. It should be noted that all interactions are considered in the presence of visible light, so the values were used for the visible light range. The nanoparticle shell is variable as it is formed by the stabilizer molecules, its thickness depends much on the nanoparticle size, stabilizer concentration and material. For the trisodium citrate the shell thickness is around 0.4-0.7 nm [39], for the calculations we used the shell thickness value of 0.5 nm and the shell dielectric constant value of 1.3 as in [40].

The virus was described as a spherical structure with a nonhomogeneous shell, the surface has some cylindric spikes (mainly glycoproteins). The radius of the inner part of the virion in the calculations is equal to 60 nm, and the whole shell thickness was taken 20 nm: 10 nm of the homogeneous layer and 10 nm is the height of the cylinders [41]. As the dielectric properties are not known exactly for the viruses, we chose the ones for DNA [42], which is the inner part, and viral proteins and glycoproteins [43], which corresponds to the shells.

The results of calculations show the energy of interaction between the nanoparticle and the virus surface depending on the distance between them. As it was shown in Fig. 3c the distance “d” changes but the relative position of the nanoparticle center and the edge of the spike are stable. The minimum of the potential indicates the physical adsorption of the nanoparticle on the virus surface, whereas the depth of the minimum indicates the energy of adsorption. The case with the deepest energy minimum is the most energetically favorable state. Consequently, comparing the potentials for different relative locations of the virus center and the spike edge, it can be seen that the deepest energy minimum and the closest position of the equilibrium system state is for the case ‘1’. Similar results may be observed for all the nanoparticles sizes considered. However, for the 20 nm nanoparticle these changes are not so obvious. Comparing potential for different sizes of the nanoparticles it can be seen that the deepest minimum is for the smallest nanoparticles. Hence, it may be supposed that the antiviral effect may be higher for the smaller nanoparticles, which was indeed observed in our work.”

The final statement of this note “However, an experimental confirmation on the molecular level that nanoparticle adsorption to the virus spike is the most energy-efficient state is a challenge that was not addressed in this preliminary communication” was added at the end of section 3.3 (page 10, line  403-406).

Figure 4 still needs improvement (text not readable).

Font size on graphs in Figure 4 has been increased.

The original version of the manuscript was checked by the MDPI editing service (the certificate available on request). The spelling check of the additions and corrections in the revised version has been carried out.

All the changes are highlighted: additions or changes - in yellow, changes in formatting - in green.

Additional changes

  1. All equations have been added as MathType objects, font was changed for Palatino Linotype 10 where needed.
  2. In some places there were no references to figure or table, just (0), we added them now, the changes are highlighted in yellow.
  3. References 38-39 were updated to 44-45 and 40 to 46: p. 11, lines 458 and 468.
  4. Conclusions: conclusion 1 was in Times New Roman, it was changed to Palatino Linotype.
  5. Acknowledgments: we would appreciate to change initial IM to IuM.